# High-Accuracy and Real-Time Fingerprint-Based Continual Learning Localization System in Dynamic Environment

**DOI:** 10.3390/s25051289

**Published:** 2025-02-20

**Authors:** Hongxiu Zhao, Wafa Njima, Xun Zhang

**Affiliations:** Department of Telecommunication Engineering, Institut Supérieur d’Electronique de Paris (ISEP), 92130 Paris, France; hongxiuzhao0705@gmail.com (H.Z.); xun.zhang@isep.fr (X.Z.)

**Keywords:** continual learning (CL), dynamic environment, fingerprinting, transfer learning (TL), rehearsal

## Abstract

In dynamic environments, localization accuracy often deteriorates due to an outdated or invalid database. Traditional approaches typically use Transfer Learning (TL) to address this issue, but TL suffers from the problem of catastrophic forgetting. This paper proposes a fingerprint-based Continual Learning (CL) localization system designed to retain old data while enhancing the accuracy for new data. The system works by rehearsing parameters in the lower network layers and reducing the training rate in the upper layers. Simulations conducted with fused data show that the proposed approach improves accuracy by 16% for new data and 29% for old data compared to TL in smaller rooms. In larger rooms, it achieves a 14% improvement for new data and a 44% improvement for old data over TL. These results demonstrate that the proposed CL approach not only enhances localization accuracy but also effectively mitigates the issue of catastrophic forgetting.

## 1. Introduction

Recently, the indoor localization problem has garnered significant attention in research, driven by the growing popularity of indoor mobile robots for transportation and artificial intelligence communication services [1,2]. Unlike outdoor localization, indoor localization presents unique challenges that classic Global Positioning System (GPS)-assisted methods cannot meet due to the stringent accuracy requirements. Especially in a highly dynamic environment, localization has to deal with the case of environmental layouts changing [3]. Meanwhile, the fingerprinting method is widely applied due to the advantages of high localization accuracy, effective linear and nonlinear models, and easy upgrade information to amend [4]. It has two phases: offline dataset establishment phase and online localization phase. In the first phase, parameters such as Received Signal Strength (RSS) and Channel State Information (CSI) are first collected from Access Points (APs)/Reference Nodes (RNs) at different known locations, and then stored in a database along with their corresponding location coordinates, therefore creating a “fingerprint” map of the environment. In the online localization phase, the measured parameters are compared to the stored fingerprints, and then by matching the current measured parameters with the closest fingerprint in the database, localization is finally achieved [5]. The key to achieving high localization accuracy with the fingerprint method lies in the real-time precision of the database. However, in a dynamic environment with multiple moving robots or services, the database is always changing [6,7]. Ref. [8] demonstrates the susceptibility of RSS to environmental variability by testing and comparing the RSS variation in six months (long term) and the one in one week (short term). According to existing research, the variation in RSS distributions at the same location over 44 days is demonstrated [9]. Consequently, the accuracy of location estimation can degrade significantly as time passes.

To solve this problem, Deep Learning (DL) methods have been used by updating the offline database training phase or online matching phase. Ref. [10] uses a Machine Learning (ML)-based feedback loop during the database online calibration step and updates the database in real time to learn the variability of the environment. The feedback data are collected by additional devices that are different from the ones used in the offline calibration. This makes continuous fingerprint calibration feasible even in the presence of different machines and Internet-of-Things (IoT) sensors that introduce variations to the electromagnetic environment. Ref. [11] explores a new paradigm of radio map construction and adaptation with few-shot relation learning. It augments the collected data and models the fundamental relationships of the neighborhood fingerprints. In this way, network updating is not necessary, decreasing the time cost. Ref. [12] applies Long Short-Term Memory (LSTM) deep network architectures to achieve localization based on collected RSS data. This study also improves the Long Short Term Memory (LSTM) architecture and conduct some comparison with other Convolutional Neural Networks  (CNNs). As a result, the proposed model achieves localization with meter-level accuracy. However, DL-based methods are still time-consuming and labor-intensive due to the collection of labeled data, which limits the applicability of fingerprint-based localization systems.

In this case, Transfer Learning (TL) is popular to solve the fingerprint adaption in a dynamic environment [13,14]. It maps features from the source domain (initial environment) and the target domain (changed environment) into the same feature space using supervised or semi-supervised learning. This enables the reuse of previously invalid labeled fingerprints in the new environment, therefore reducing the reliance on labeled data in the new environment. Ref. [6] improves the traditional fingerprint method in both offline establishment and online localization stages. It applies the Domain-Adversarial Neural Network (DANN) [15] based on TL in the offline stage to construct the database using only unlabeled data and the Passive Aggressive (PA) algorithm [16] in the online stage to track the dynamic characteristics of the environment and calibrate the entire localization system. Transfer Learning methods such as zero-shoot learning and one-shoot learning focus on excelling at new tasks but fail to prevent the catastrophic forgetting of previously learned tasks or other new tasks in the same domain [17]. In this case, as the fingerprinting database changes over time and the new task is learned, the old information cannot achieve high-precision localization when applied to the new model. For example, let us assume the following: (1) the old database is established at time t−1; (2) a new model $M_{t}$ is created based on new information at time *t*; (3) catastrophic forgetting occurs during this TL process; (4) the database at time t+1 is identical to the one at time t−1; and (5) model $M_{t+1}$ is trained using the same TL approach and the database at time t+1. Consequently, the localization accuracy at time t+1 is lower than that at time t−1, illustrating the effects of catastrophic forgetting.

Continual Learning (CL) is applied to deal with the catastrophic forgetting by using only new data. Usually, it can be categorized into (1) regularization-based methods [18,19], which work by penalizing changes in the critical weights associated with previous tasks or introducing orthogonality constraints in the training objectives, thereby reducing the risk of the model forgetting prior knowledge; (2) modular-based methods [20], which expand the model’s capacity by assigning new parameters to each task; and (3) replay-based methods [21], which store samples from old scenes in a fixed-size buffer or using generative models to reconstruct and reproduce images of old scenarios, and therefore experience replay can be implemented. In theory, CL approaches could be leveraged not only to manage forgetting but also to decrease training complexity [22]. In fact, regularization-based methods require the retention of past gradients or feature maps, akin to replay-based methods, to prevent forgetting and ensure the continuity of learning. In contrast, modular-based methods involve increasing the model’s capacity by adding new parameters for each task, which can complicate the model’s structure. When the model size is fixed, experience replay-based methods have been shown to outperform regularization-based methods in preserving previously learned knowledge. Moreover, integrating regularization and experience replay approaches has been demonstrated with more effective results [23]. Therefore, in this paper, we apply a replay-based method to fulfill localization in a dynamic environment. Until now, only few studies have applied CL in the wireless localization field. Ref. [22] proposed a novel real-time CL method based on “Latent Replay” policy. It integrates old data at a selected intermediate layer, rather than the input layer, based on the desired balance between accuracy and efficiency. Meanwhile, it keeps the layers below the selected layer frozen not only to decrease the time cost but also to save the old information. A year later, the same authors applied this novel method to localize Unmanned Aerial Vehicles (UVAs) outdoors using a multi-camera and assisted by the GPS information [24]. In this paper, mini-batches are generated first, then Convolutional Neural Network (CNN)-based continual training using the old created mini-batches, and finally the available model is used to classify the current image, providing a discrete label that can be mapped to a GPS coordinate. This is the first work to localize the target based on the replay-based CL. Later, the authors improved the localization system by using a multi-model so that localization can be accomplished even when GPS cannot work [25]. At the same time, another research team also applied CL to accomplish localization. A robust baseline was proposed in [26] that relies on storing and replaying images from a fixed memory called buffer. Additionally, the study introduces a novel sampling method called Buff-Coverage Score (Buff-CS), which adapts existing sampling strategies within the buffering process to address catastrophic forgetting. These studies all utilize images as their input. In image-based CL localization methods, the lower layers of a CNN process fundamental information with lower complexity and faster processing speeds compared to the higher layers [24]. This behavior contrasts that observed in data-driven Neural Network (NN) models, such as those commonly employed in RSS-based fingerprinting for indoor localization.

Therefore, in this paper, we propose a novel data-based CL method for real-time indoor localization. It uses replay policy and reintroduces previously learned information not directly from the input layer but through intermediate layers, allowing for more refined feature extraction and processing. Meanwhile, it improves localization accuracy by leveraging features extracted from multiple lower layers rather than depending on a single, predetermined layer. By utilizing a broader range of feature representations such as some weights and bias, it ensures a more comprehensive understanding of the data, leading to more robust and precise localization results. Furthermore, it reduces training complexity and time cost by freezing the upper layers, which is a different approach from that used in image-based CL methods. Overall, the contributions in this paper are summarized as follows:This paper is the first to apply a data-based CL method for indoor localization, whereas previous approaches to CL in indoor localization have all been image-based.This paper enhances indoor localization accuracy and avoids catastrophic forgetting by replaying some partial features across multiple layers, rather than relying on a single layer.It accelerates training time by freezing the upper layers, as they involve higher complexity.The proposed system enables real-time localization thanks to the robustness of the trained model, which incorporates information from multiple time periods.

To present this work, our paper is organized as follows: Section 2 provides an overview of the system model, including data generation and the NN structure. Section 3 describes the proposed data-based CL architecture for high-accuracy and real-time localization. The simulations and the analysis of the results are presented in Section 4. Finally, Section 5 provides the conclusions and outlines future work.

## 2. System Model

In this section, we will present the data generation process along with the corresponding NN structures. We will detail how the data are collected, prepared, and utilized, and then describe the design and architecture of the NN models used. This includes a comprehensive overview of the network layers, their configurations, and how they integrate with the data generation pipeline to support effective training and localization.

### 2.1. Fusion Data Generation

As demonstrated in our previous work, data fusion effectively addresses the issue of data loss and significantly improves localization accuracy in dynamic environments, such as industrial factories. It combines information from multiple sources (e.g., Access Points (APs)/Reference Nodes (RNs)) to mitigate the impact of missing data and enhance the overall precision of localization systems. In addition, Optical Wireless Channel (OWC) localization technology has garnered considerable attention due to its advantages in information security, cost-effectiveness, and minimal signal interference [27]. OWC offers a secure and reliable alternative to traditional wireless communication methods, making it particularly suitable for environments where data integrity is crucial. Simultaneously, Wi-Fi technology remains a popular choice due to its cost efficiency and widespread standardization through protocols such as IEEE 802.11ax [28,29]. The ubiquity of Wi-Fi makes it an accessible and practical solution for many localization applications, contributing to its continued use in various settings. Given the complementary strengths of OWC and Wi-Fi technologies, we propose a system that integrates OWC-RSS and Wi-Fi-RSS data as inputs. By combining these data sources, we aim to leverage their individual advantages to enhance the robustness and accuracy of the localization system.

Typically, a localization system comprises three main roles: APs, RNs, and User Equipment (UE) [30]. APs serve as fixed infrastructure points that transmit and receive signals, providing reference information for localization calculations. RNs are positioned at known locations to help determine the locations of UE through localization methods such as triangulation. UE are the devices whose locations are determined, such as mobile phones or other user devices. In addition to these primary components, External Interferences (EIs) may also be present during system operations, especially in a dynamic environment. Although EIs do not communicate directly with the other components, they can affect the accuracy and reliability of the localization system by introducing signal distortions or other disturbances. Therefore, we first consider the dynamic indoor localization environment shown as Figure 1. The room has dimensions of *L* in length, *H* in height, and *W* in width. Inside, different devices are linked through different transmitters through OWC or Wi-Fi connections. In dynamic environments, the presence of EIs can impact signal transmission. As a result, UE may not consistently receive complete RSS data, leading to partial data loss. These incomplete data can significantly affect the accuracy and reliability of the localization process.

To model the dynamic environment, we first fix the dimensions of the room as follows: *L* = 5 m; *W* = 5 m; and *H* = 5 m. We then consider two primary variables: the proportion of missing data and the noise levels in both OWC and Wi-Fi channels. When the missing data proportion is set to 20%, we examine two different noise scenarios: (1) Dataset 1, where the Signal-to-Noise Ratio (SNR) in OWC channels is 40 dB, and the standardization value in Wi-Fi channels is 2, and (2) Dataset 2, where the SNR in OWC channels is increased to 50 dB, while the standardization value in Wi-Fi channels is set to 5. Similarly, when the missing data proportion is increased to 50%, we collect two additional datasets under the same noise conditions: (1) Dataset 3, which corresponds to an SNR of 40 dB in OWC channels and a standardization value of 2 in Wi-Fi channels, and (2) Dataset 4, which corresponds to an SNR of 50 dB in OWC channels and a standardization value of 5 in Wi-Fi channels. Similarly, if we set the dimensions at *L* = 20 m, *W* = 20 m, and *H* = 10 m, we can define Datasets 5, 6, 7, and 8 under the same noise and missing data conditions. These are summarized in Table 1. For each dataset, we have 5 measurements. In each environment configuration, 4 OWC-APs, arranged in a quadrilateral shape, are placed on the ceiling and 10 Wi-Fi-APs are positioned at a height of 10 cm, arranged in multiple quadrilateral patterns throughout the room.

### 2.2. CL Rehearsal Policy Overview

Basic rehearsal policy is a classic CL method using a stream of data and artificial intelligence networks [31]. It involves storing a subset of previously encountered data and periodically incorporating it into the training of new tasks. The goal is to maintain a balance between new and old knowledge, ensuring that the model retains essential information over time. Two key factors are vital: (1) external memory M, which stores past data; the size and composition of this memory are crucial for balancing the impact of old and new data; and (2) rehearsal frequency, which leads to batch size *B* and proportion *P* which is the ratio of old to new data during training, playing a significant role in the model’s ability to retain prior knowledge while learning new tasks.

The pseudocode in Algorithm 1 outlines the strategy to manage external memory M during training, ensuring that it does not overflow and that the contribution from different batches *b* is balanced without enforcing the balance of the classes.
**Algorithm 1** Basic CL rehearsal pipeline  1:**Input:** Training data D, Batch size *B*, External memory M  2:**Output:** Updated model with parameters θ  3:Initialize model parameters θ  4:Initialize the external memory M  5:**for** each epoch **do**  6:      Shuffle training data D  7:      **for** each batch *b* in D **do**  8:            **if** first batch **then**  9:                 Train model based on batch *b*10:            **else**11:                 Train model based on batch *b* and external memory M12:                 Sample a random subset of data $S_{b}$ from batch *b* based on the portion *P*13:                 Sample a random subset $S_{m}$ which has the same size of $S_{b}$ from external memory M14:                 Discard $S_{m}$ from external memory M15:                 Add $S_{b}$ to external memory M16:            **end if**17:      **end for**18:**end for**19:**Return** Updated model with parameters θ

It should be noticed that this differs from traditional TL methods such as fine-tuning which is used to adapt pre-trained models to new tasks or datasets. In this work, we use fine-tuning as an example to represent the TL method. Fine-tuning leverages the knowledge acquired by a model during its initial training phase and refines it to suit the specific requirements of a new, often smaller, dataset. The overall pipeline is shown in Algorithm 2. This process is particularly useful in scenarios where training a model from scratch would be computationally expensive or when the available data are insufficient to train a model effectively from the ground up. Furthermore, the main difference can be observed in the pseudocode provided below.
**Algorithm 2** Basic fine-tuning Transfer Learning pipeline  1:**Input:** Pre-trained model *M*, New training data D, Batch size *B*, Learning rate η  2:**Output:** Updated model with parameters θ  3:**for** each epoch **do**  4:      Shuffle training data D  5:      **for** each batch *b* in *D* **do**  6:            Compute predictions $\hat{y}$ in Forward pass  7:            Compute loss $L(\hat{y},y)$  8:            3: Compute gradients of the loss for trainable parameters in Back-propagate pass  9:            Update parameters with learning rate η10:      **end for**11:**end for**12:**Return** Updated model with parameters θ

It should be noted that in this basic fine-tuning process, no old information is needed except at the beginning of training, which may lead to catastrophic forgetting.

## 3. Proposed Architecture

As mentioned above, we propose an improved rehearsal-based CL system for real-time localization. The proposed system is designed with two primary objectives: preventing the catastrophic forgetting of previous data and ensuring high localization accuracy for new data. The first objective is addressed through the rehearsal strategy detailed in Section 2. Traditional rehearsal-based CL methods typically focus on replaying data at the input layer, while the limited existing research on CL for localization emphasizes rehearsals at the chosen intermediate layer. To further enhance localization accuracy, our system extends the rehearsal policy to multiple intermediate layers, rather than relying on a single layer. In addition, to reduce computational complexity, the processing speed in the upper layers—where most of the computational load occurs—is deliberately slowed down. The second objective, maintaining localization accuracy for new data, is achieved by initializing the parameters in the lower layers using a partially trained NN based on the new dataset, which ensures that the system remains efficient while effectively integrating new information. The whole process will be explained further. First, at time *t*, a trained $NN_{t}$ based on old data is determined. The NN is structured into three main types of layers: the input layer, which receives the initial data; hidden layers, where complex computations and feature extractions occur; and the output layer, which produces the final result or prediction. Each layer comprises multiple neurons, which are the basic units that process the information. In a traditional fully connected NN, also known as a dense network, every neuron in one layer is linked to every neuron in the following layer. These connections allow for the full transfer of information from one layer to the next. Specifically, the output from each neuron in a layer is passed through an activation function, which introduces non-linearity into the model, and then this processed output becomes the input for the neurons in the subsequent layer. For the input layer, let Input represent the input data (which in this case are the fusion data with missing values), and the output is given by $\mathrm{output}=f_{1}\left(\mathrm{Input}\right)$, where $f_{1}\left(\cdot \right)$ denotes the activation function. For the hidden layer, the output is calculated as $\mathrm{output}=f_{h}\left(\mathrm{input}\right)=\mathrm{active}\left(W\cdot \mathrm{input}+b\right)$, where *W* refers to the weights connected to each neuron, and *b* is the bias term. An example of this is seen in the first hidden layer, as illustrated in Figure 2. The output layer functions similarly to the hidden layers. This whole process is generally as shown in the upper left corner of Figure 2. The dense connections and layered structure enables the NN to learn and represent complex patterns within the data.

At time t+1, new data arrive, and the corresponding $NN_{t+1}$ is trained following the traditional neural network approach. During this process, the external memory is refreshed. Initially, selected layers of $NN_{t+1}$ have their parameters extracted. The CL network is then trained by initializing its parameters, including weights and biases, based on the pre-trained model $NN_{t}$. In this stage, the lower layers are updated by integrating the parameters stored in the external memory. To manage the high computational complexity of the upper layers, we reduce the training speed specifically for these layers. This strategy prevents the model from being overwhelmed by the complex adjustments required in the upper layers while still incorporating new information effectively. As a result, we obtain the updated CL model, denoted as $CL_{t+1}$, which integrates both the new data and the retained knowledge from previous training.

The complete pipeline of the proposed CL system is outlined in Algorithm 3. In this paper, key functions are applied. Gradient descent function $\nabla_{\theta }L\left(\theta \right)$ is the gradient of function L(θ) with respect to θ. $\theta {|}_{L}$ are parameters chosen from *L* layers in one network. Also, key parameters in the rehearsal process are utilized. Training data D represent the raw data RSS and the corresponding real locations for training to generate the CL model. Parameters θ include weights and bias in one network. Batch size *B* is the number of receivers used per iteration. The learning rate η controls the step size taken in the direction of the gradient during each update in gradient-based optimization. Layers for fine-tuning *L* indicate the layers that are updated during the fine-tuning process. External memory M refers to a storage mechanism used to retain previous and old RSS data. Two networks work in the whole pipeline: the old network NN and the new network ${\mathrm{NN}}^{\prime }$. NN is trained based on the old RSS data and ${\mathrm{NN}}^{\prime }$ is trained by the new RSS data.
**Algorithm 3** Proposed Continual Learning rehearsal pipeline for fingerprint-based indoor real-time localization  1:**Input:** Training data D, Batch size *B*, Learning rate η, Layers for fine-tune *L*, External memory M  2:**Output:** Updated model with parameters θ  3:Initialize model parameters θ  4:Initialize the external memory M  5:**for** each epoch **do**  6:      Shuffle training data D  7:      **for** each batch *b* in D **do**  8:            **if** first batch **then**  9:                  Train model based on batch *b*10:                  Update model parameters:$$\theta \leftarrow \theta -\eta \nabla_{\theta }L\left(b,\theta \right)$$
where L is the loss function.11:            **else**12:                   Train model based on batch *b* for the new network ${\mathrm{NN}}^{\prime }$13:                   Sample parameters $\theta_{L}^{\prime }$ from selected layers *L* of ${\mathrm{NN}}^{\prime }$:$$\theta_{L}^{\prime }=\theta^{\prime }{|}_{L}$$14:                   Update the external memory M with sampled parameters $\theta_{L}^{\prime }$:$$M\leftarrow M\cup \{\theta_{L}^{\prime }\}$$15:                   Incorporate data stored in M into the previous network NN:16:                   - Randomly discard parameters $\theta_{L}$ in selected layers of NN:$$\theta_{L}\leftarrow M\;\mathrm{or}\;\mathrm{randomly}\;\mathrm{discarded}\;\mathrm{data}$$17:                   - Fuse parameters in selected layers *L* from external memory M:$$\theta_{L}\leftarrow \theta_{L}\cup M$$18:                   Perform fine-tuning on the previous NN by minimizing the loss function:$$\theta \leftarrow \theta -\eta \nabla_{\theta }L\left(b,\theta \right)$$19:                   Reduce the training speed for upper layers *L* by updating learning rate η:$$\eta_{L}\leftarrow \eta \times \alpha$$
where α∈[0,1] is a reduction factor.20:            **end if**21:      **end for**22:**end for**23:**Return** Updated model with parameters θ

## 4. Simulation Analysis

In this section, we present the performance evaluation of the proposed CL system for real-time localization and compare it with traditional fully connected feed-forward DL systems, which employ Levenberg–Marquardt optimization with a learning rate of 0.01, 100 epochs, and a batch size of 100. Additionally, we compare it with the TL system, using fine-tuning as a benchmark. Therefore, we have four models: $NN_{1}$, which is traditionally trained using the old dataset; $NN_{2}$, which is traditionally trained using the new dataset; a fine-tuning model trained based on a basic TL system; and finally, our proposed CL model. In the proposed CL model, we aim to identify the optimal configuration by testing various data rehearsal proportions, specifically 0%, 20%, 50%, 80%, and 100%. In training neural networks, various structures are explored to identify the optimal configuration. In this context, TL is implemented using fine-tuning with a consistent network structure. Similarly, CL is tested with different rehearsal memory sizes, but also using the same network structure.

We have two primary goals as introduced in Section 3: preventing the catastrophic forgetting of previous data and ensuring high localization accuracy for new data. In this case, the accuracy of the fine-tuning model on the old dataset is expected to be lower than that of the CL model on the same dataset. Additionally, the accuracy of $NN_{2}$ on the new dataset is expected to be lower than that of the CL model on the same dataset.

Four verification tests are conducted to validate the system’s effectiveness in dynamic environments, considering not only variations in missing data proportions but also changes in noise levels. Datasets are generated as firstly introduced in Section 2. In each dataset, we take 70% data for training process and 30% data for validation or testing. In this study, for the rehearsal policy, we rehearse partial neurons from the first three hidden layers of a network with five hidden layers, each containing ten neurons. In Table 2, we present the used NN architectures where $L_{i}\left(.\right)$ is the number of neurons in the $i^{th}$ hidden layer, and to guarantee the adaption between NNs, we set the same architecture for each NN. Here, we choose $L_{1},L_{2},L_{3}$ as our rehearsal layers.

### 4.1. Verification When Missing Data Proportion Changes over Time

To simulate a dynamic environment, we utilize two datasets to represent the data at time *t* and time t+1, respectively. To assess the system’s ability to prevent catastrophic forgetting, we choose a dataset with a higher missing data proportion for time *t* and a dataset with a lower missing data proportion for time t+1. Specifically, in the smaller room, Dataset 1 represents the data at time t+1, while Dataset 3 represents the data at time *t*, forming a comparison pair. Similarly, Dataset 2 and Dataset 4 form another pair. Here, we will present the results of the first comparison.

The Cumulative Distribution Function (CDF) of the localization error for various networks in the small room is shown in Figure 3. To provide a comprehensive evaluation, the results are presented separately for each dataset, highlighting the network’s performance with the new data Dataset 1 at time t+1 and the old data Dataset 3 at time *t*.

As shown in Figure 3a, for the new dataset, CL networks, irrespective of the rehearsal memory size, consistently exhibit faster convergence compared to the DL network trained solely on the new dataset. This indicates that the proposed CL approach more effectively integrates new information while retaining prior knowledge. A particularly noteworthy observation is that in the case of 80% receivers, the localization error for the CL network with RM=20% is significantly lower than that of the fine-tuning network. These findings are visually represented in Figure 3a, further validating the advantage of using a CL approach in dynamic environments. Meanwhile, when evaluating the performance on the old dataset, as shown in Figure 3b, the DL network still outperforms the other models, achieving the best results. However, the proposed CL network with RM=20% closely follows, ranking as the second-best performer. This outcome further reinforces the effectiveness of the CL approach in balancing new and old knowledge. Furthermore, it should be noted that the fine-tuning network exhibits the slowest convergence rate, underscoring the limitations of the fine-tuning approach in maintaining accuracy on previously learned data and further emphasizing the advantage of the CL network in mitigating catastrophic forgetting. The same analysis and conclusion can also be verified with the result when counting for 80% receivers. Therefore, the proposed CL approach with 20% data rehearsal not only improves the localization accuracy for the new data compared to the fine-tuning approach, but also avoids catastrophic forgetting for the previous dataset. Here, we provide the concrete localization Mean Squared Error (MSE) results generated by the DL method, the TL method with fine-tuning, and the proposed CL approach with a 20% rehearsal proportion, as exhibited in Table 3. In this table, as introduced above, the old dataset is the one with 50% data missing, which is Dataset 3, and the new dataset is the one with 20% data missing, which is Dataset 1.

In Table 3, we observe that for the new dataset, the proposed CL approach achieves a localization MSE of 0.616 m, which is an improvement of 16% over the fine-tuning approach and 66% over the DL approach. This confirms the high localization accuracy of the proposed CL approach. For the old dataset, the error of the proposed CL approach is 0.681 m, a 47% improvement compared to the fine-tuning approach, further validating its ability to mitigate catastrophic forgetting.

Similarly, in the larger room, where Dataset 7 represents the old dataset and Dataset 5 represents the new dataset, the proposed CL approach demonstrates superior localization accuracy compared to the TL approach. It also effectively mitigates catastrophic forgetting, as evidenced by an 80% rehearsal proportion. As shown in Table 4, it is observed that with an 80% rehearsal proportion, the proposed CL approach achieves a localization MSE of 2.551 m for the new dataset, which represents a 19% improvement over the fine-tuning approach and a 35% improvement over the DL approach, confirming its high localization accuracy. For the old dataset, the proposed CL approach yields an error of 6.297 m, reflecting a 50% improvement compared to the fine-tuning approach, further validating its ability to mitigate catastrophic forgetting.

### 4.2. Verification When Noise Varies over Time

We also verify the proposed CL approach in a dynamic environment with noise variation over time. In this scenario, within the smaller room, Dataset 2 serves as the old dataset and Dataset 1 as the new one, simulating a dynamic environment. Additionally, Dataset 4 and Dataset 3 can form another comparison pair. Here, we present the results from the first dataset pair.

As shown in Figure 4, similar observations are made as in Figure 3, with the proposed CL approach demonstrating the best performance among the presented rehearsal proportions at 20%. For the new dataset, CL networks convergence faster compared to the DL network trained exclusively on the new dataset, regardless of the rehearsal memory size. This highlights the CL approach’s superior ability to integrate new information while preserving prior knowledge. Notably, for 80% receivers, the localization error for the CL network with RM=20% is lower than that of the fine-tuning network. These findings, shown in Figure 4a, underscore the advantages of the CL approach in dynamic environments for the new dataset. It can be noticed that the results from fine-tuning and CL with RM=0%, RM=20%, RM=50%, RM=80%, and RM=100% are very similar. This is because in conditions where only small changes occur in the environment, indicating that the rehearsal data are similar to the previous ones and further implying that the new dataset has minimal differences from the old one, fine-tuning and CL may achieve similar accuracy performance. For the old dataset, as shown in Figure 4b, the proposed CL approach with 20% rehearsal proportion performs higher localization accuracy and also avoids catastrophic forgetting. The MSE results are shown in Table 5.

As presented in Table 5, the proposed CL approach with a 20% rehearsal proportion achieved a localization MSE of 0.048 m for the new dataset, which represents a 29% improvement compared to the fine-tuning approach and a 26% improvement compared to the DL approach. For the old dataset, the proposed approach also showed a 29% improvement over the fine-tuning approach, verifying its ability to avoid catastrophic forgetting.

For the larger room, similar conclusions are observed with an 80% rehearsal proportion. As shown in Table 6, the proposed approach demonstrates higher localization accuracy, improved by 14% compared to the fine-tuning approach for the new dataset. Additionally, it mitigates catastrophic forgetting by achieving a 44% improvement compared to the fine-tuning approach for the old dataset.

## 5. Conclusions

In this study, we proposed a fingerprint-based CL localization system using fused data, which further enhances the accuracy of the localization system. By reducing the learning rate for higher-level layers, we addressed the issue of catastrophic forgetting, thereby improving the retention of old information. At the same time, information from partial lower-layer neurons is preserved to effectively capture new information. The reduction in the learning rate for higher layers also improves the system’s real-time processing performance. This research is the first one using fingerprinting for CL localization in a dynamic environment. The current study evaluated the performance of this localization method under a single rehearsal policy (rehearsing neurons from the first three hidden layers). Future work will expand this to explore various hidden layer extraction schemes to further optimize performance.

## Figures and Tables

**Figure 1 sensors-25-01289-f001:**
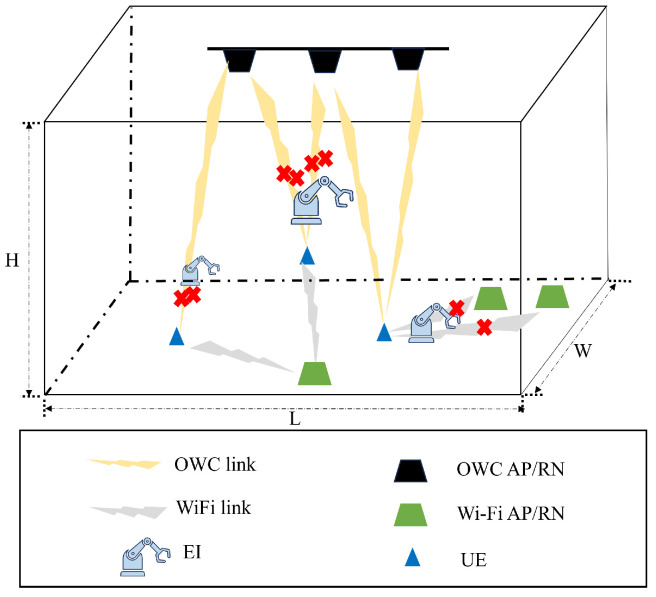
System model in an dynamic localization environment with data missing.

**Figure 2 sensors-25-01289-f002:**
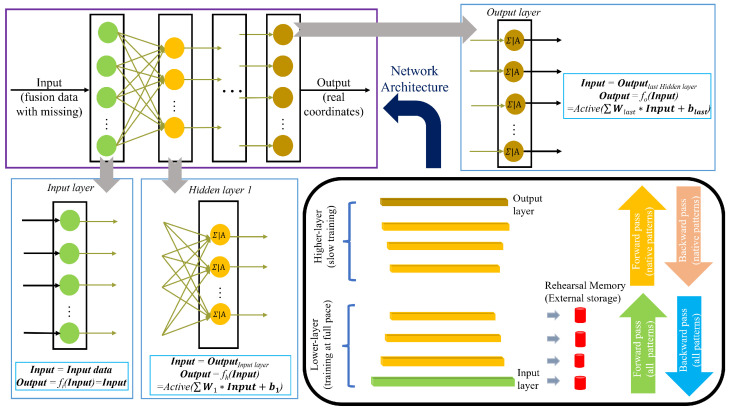
Pipeline of the proposed system.

**Figure 3 sensors-25-01289-f003:**
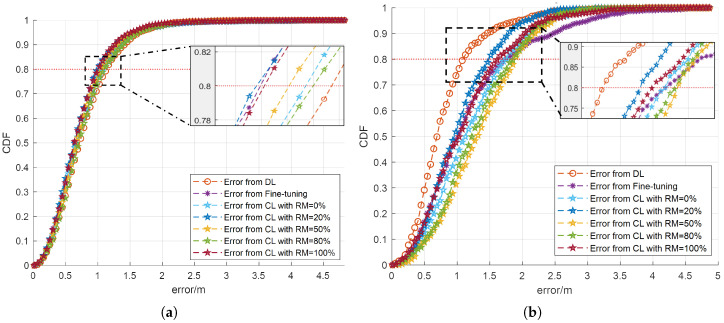
The CDF of the localization error across different networks, evaluated using both new and old datasets, in a small dynamic room where the missing data proportion varies. (**a**) The CDF of the localization error generated using the new dataset (**b**) The CDF of the localization error generated using the old dataset.

**Figure 4 sensors-25-01289-f004:**
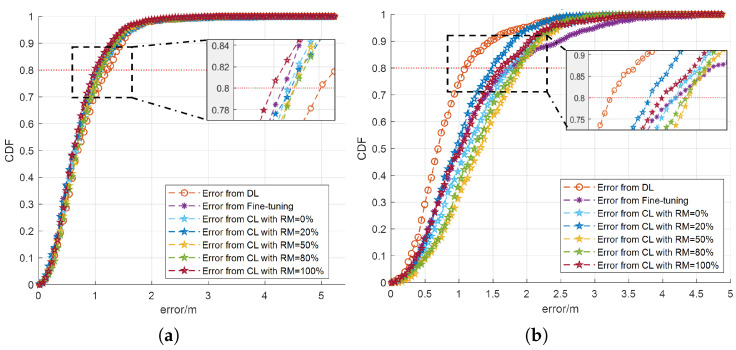
The CDF of the localization error across different networks, evaluated using both new and old datasets, in a small dynamic room where the noise varies. (**a**) The CDF of the localization error generated using the new dataset (**b**) The CDF of the localization error generated using the old dataset.

**Table 1 sensors-25-01289-t001:** Datasets generated in different cases where features are indicated by ✔.

	Room Configuration (m * m * m)		Data Missing (%)		Noise Condition	
	5 * 5 * 5	20 * 20 * 20	20	50	OWC-SNR = 40 dB	OWC-SNR = 50 dB
					Wi-Fi-std = 2	Wi-Fi-std = 5
Dataset 1	✔		✔		✔	
Dataset 2	✔		✔			✔
Dataset 3	✔			✔	✔	
Dataset 4	✔			✔		✔
Dataset 5		✔	✔		✔	
Dataset 6		✔	✔			✔
Dataset 7		✔		✔	✔	
Dataset 8		✔		✔		✔

**Table 2 sensors-25-01289-t002:** Architectures of different used models.

Used NN	Architecture
$NN_{1}$	$L_{1}\left(10\right)$, $L_{2}\left(10\right)$, $L_{3}\left(10\right)$, $L_{4}\left(10\right)$, $L_{5}\left(10\right)$
$NN_{2}$	$L_{1}\left(10\right)$, $L_{2}\left(10\right)$, $L_{3}\left(10\right)$, $L_{4}\left(10\right)$, $L_{5}\left(10\right)$
Fine-tuning	$L_{1}\left(10\right)$, $L_{2}\left(10\right)$, $L_{3}\left(10\right)$, $L_{4}\left(10\right)$, $L_{5}\left(10\right)$
Proposed CL	$L_{1}\left(10\right)$, $L_{2}\left(10\right)$, $L_{3}\left(10\right)$, $L_{4}\left(10\right)$, $L_{5}\left(10\right)$

**Table 3 sensors-25-01289-t003:** Localization MSE (m) based on different networks. The rehearsal portion is 20% in the proposed CL system in the first room configuration under the case that the missing data proportion changes over time. The “Old Dataset” is Dataset 3 with 50% missing data proportion and the “New Dataset” is Dataset 1 with 20% missing data proportion.

	${\mathrm{NN}}_{1}$	${\mathrm{NN}}_{2}$	Fine-Tuning	Proposed CL
Old Dataset	0.481	-	1.287	0.681
New Dataset	-	1.815	0.733	0.616

**Table 4 sensors-25-01289-t004:** Localization MSE (m) based on different networks. The rehearsal portion is 80% in the proposed CL system in the second room configuration under the case that the missing data proportion changes over time. The “Old Dataset” is Dataset 7 with 50% missing data proportion and the “New Dataset” is Dataset 5 with 20% missing data proportion.

	${\mathrm{NN}}_{1}$	${\mathrm{NN}}_{2}$	Fine-Tuning	Proposed CL
Old Dataset	4.151	-	12.539	6.297
New Dataset	-	3.903	3.143	2.551

**Table 5 sensors-25-01289-t005:** Localization MSE (m) based on different networks. The rehearsal proportion is 20% in the proposed CL system in the first room configuration under the case that the noise changes over
time. The “Old Dataset” is Dataset 2 and the “New Dataset” is Dataset 1, which is in a different noise condition.

	${\mathrm{NN}}_{1}$	${\mathrm{NN}}_{2}$	Fine-Tuning	Proposed CL
Old Dataset	0.454	-	1.111	0.785
New Dataset	-	0.552	0.576	0.408

**Table 6 sensors-25-01289-t006:** Localization MSE (m) based on different networks. The rehearsal proportion is 80% in the proposed CL system in the second room configuration under the case that the noise changes over
time. The “Old Dataset” is Dataset 6 and the “New Dataset” is Dataset 5, which is in a different noise condition.

	${\mathrm{NN}}_{1}$	${\mathrm{NN}}_{2}$	Fine-Tuning	Proposed CL
Old Dataset	3.201	-	11.661	6.503
New Dataset	-	8.534	2.932	2.519

## Data Availability

The original contributions presented in this study are included in the article. Further inquiries can be directed to the corresponding author.

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
