# Peer review of "High-Accuracy and Real-Time Fingerprint-Based Continual Learning Localization System in Dynamic Environment"

_sensors, 2025, doi:10.3390/s25051289_

Round 1

Reviewer 1 Report

Comments and Suggestions for Authors

This paper proposes a fingerprint-based Continual Learning (CL) localization system designed to retain old data while enhancing the accuracy for new data. The authors conduct the experiments to show the improvements of the proposed method with the existing schemes in effectiveness and accuracy. However, there are some issues need to be clarified or addressed to make this manuscript to meet the publication requirements.

1. The introduction and literature review section primarily rely on older references and lacks sufficient engagement with recent developments in the last three years.

2. The proposed Algorithm 3 lacks detailed mathematical formulations, parameter definitions, and tuning mechanisms, which are essential for reproducibility and validation of the approach.

3. In Tables 3–6, the manuscript does not clearly explain the differences between the old and new datasets. A detailed description of their characteristics would help readers understand the context and significance of using both datasets.

4. Tables 3–6 show that NN1 and NN2 models are not consistently validated on both the old and new datasets. To ensure a comprehensive evaluation, it would be beneficial to include results for NN1 and NN2 on both datasets, allowing for a direct comparison and a clearer understanding of their strengths and limitations.

5. The experiment should include some comparisons between the results obtained and the performance of the techniques/solutions from the related research to evaluate with more perspective the obtained performance.

Author Response

Review 1:

This paper proposes a fingerprint-based Continual Learning (CL) localization system designed to retain old data while enhancing the accuracy for new data. The authors conduct the experiments to show the improvements of the proposed method with the existing schemes in effectiveness and accuracy. However, there are some issues need to be clarified or addressed to make this manuscript to meet the publication requirements.

Comment 1. The introduction and literature review section primarily rely on older references and lacks sufficient engagement with recent developments in the last three years.

Answer:

Thank you for pointing it out and I agree with your idea. Therefore, I did the recent researches further.

Modification in pages 1-4: I modified and updated the related researches in the Chapter 1, such as [5,6,7,9.11,14,17,18,19,20].

Comment 2. The proposed Algorithm 3 lacks detailed mathematical formulations, parameter definitions, and tuning mechanisms, which are essential for reproducibility and validation of the approach.

Answer:

Thank you for this suggestion and to make this paper more clear about the proposed algorithm, I made more formulations, definitions and tuning mechanisms in Chapter 3.

Modification under lines 262-273 in page 8/14: The complete pipeline of the proposed CL system is outlined as follows. In this paper, key functions are applied. Gradient descent function ∇θ L(θ) is the gradient of function L(θ) with respect to θ. θ |L are parameters chosen from L layers in one network. Also key parameters in the rehearsal process are utilized. Training data D represents the raw data RSS and according real locations for training to generate the CL model. Parameters θ include weights and bias in one networks. Batch size B is the number of receivers used per iteration. Learning rate η controls the step size taken in the direction of the gradient during each update in gradient-based optimization. Layers for fine-tune L indicates the layers that are updated during the fine-tuning process. External memory M refers to a storage mechanism used to retain previous and old RSS data. Two networks work in the whole pipeline: the old network N N and the new network N N ′. N N is trained based on the old RSS data and N N ′ is trained by the new RSS data.

Comment 3. In Tables 3–6, the manuscript does not clearly explain the differences between the old and new datasets. A detailed description of their characteristics would help readers understand the context and significance of using both datasets.

Answer:

This suggestion is advisable and helps readers understand. Therefore, I made more clarifications about these table.

  • For Table 3 and Figure 3, more information about the old and new datasets are available in the description and title in Chapter 4.

Modification in description under lines 335-337 in paper 11/14 : In this table, as introduced above, old dataset is the data with 50 \% data missing which is Dataset 3 and new dataset is the one with 20 \% data missing which is Dataset 1;

Modification in title in Table 3: Localization MSE (m) based on different networks, meanwhile the rehearsal portion is 20 % in proposed CL system in the first room configuration, under the case that data missing proportion changes over time where old dataset is Dataset 3 with 50 % data missing proportion and new dataset is Dataset 1 with 20 % data missing proportion.

  • For Table 4, more information about the old and new datasets are available in the title in Chapter 4.

Modification in title in Table 4: Localization MSE (m) based on different networks, meanwhile the rehearsal portion is 80 % in proposed CL system in the second room configuration, under the case that data missing proportion changes over time where old dataset is Dataset 7 with 50 % data missing proportion and new dataset is Dataset 5 with 20 % data missing proportion.

  • For Table 5, more information about the old and new datasets are available in the title in Chapter 4.

Modification in title in Table 5: Localization MSE (m) based on different networks, meanwhile the rehearsal portion is 80 % in proposed CL system in the second room configuration, under the case that data missing proportion changes over time where old dataset is Dataset 7 with 50 % data missing proportion and new dataset is Dataset 5 with 20 % data missing proportion.

  • For table 6, more information about the old and new datasets are available in the title in Chapter 4.

Modification in title in Table 6: Localization MSE (m) based on different networks, meanwhile the rehearsal portion is 20 % in proposed CL system in the first room configuration, under the case that the noise changes over time where old dataset is Dataset 6 and new dataset is Dataset 5 that is in different noise condition.

Comment 4. Tables 3–6 show that NN1 and NN2 models are not consistently validated on both the old and new datasets. To ensure a comprehensive evaluation, it would be beneficial to include results for NN1 and NN2 on both datasets, allowing for a direct comparison and a clearer understanding of their strengths and limitations.

Answer:

Thank you for this question. Initially, the purpose of this paper is to address the problem of catastrophic forgetting and improve localization accuracy on new data.

To verify the improvement in catastrophic forgetting, it is sufficient to compare the results of NN1, fine-tuning, and the proposed algorithm on the old data.

To verify the enhancement in localization accuracy, it is sufficient to compare the results of NN2, fine-tuning, and the proposed algorithm on the new data.

Additionally, due to the sequential training process of NN1 and NN2 in the pipeline, old data is never input into NN2. Therefore, evaluating NN2’s performance on old data is not meaningful. Similarly, new data is never input into NN1, so evaluating NN1’s performance on new data is also not meaningful.

As a result, this paper does not consider the results of old data input into NN2 or new data input into NN1.

Comment 5. The experiment should include some comparisons between the results obtained and the performance of the techniques/solutions from the related research to evaluate with more perspective the obtained performance.

Answer:

Thank you for this comment which is very meaningful and insightful. Compare to other researches, this work is the first to apply continuous learning to RSS fingerprint-based indoor wireless localization. There are no directly comparable baseline results available in the literature. For example, for the team studying continual learning conducted by Gabriele Graffieti and Lorenzo Pellegrini, they only focus on the outdoor localization [1] or moving target [2], and image processing [1,2,3]. Meanwhile, for the professional team working about continual learning localization conducted by Shuzhe Wang and Zakaria Laskar, they only focus on images not RSS data.

[1] Pellegrini L, Graffieti G, Lomonaco V, et al. Latent replay for real-time continual learning[C]//2020 IEEE/RSJ International Conference on Intelligent Robots and Systems (IROS). IEEE, 2020: 10203-10209.

[2] Pellegrini L, Lomonaco V, Graffieti G, et al. Continual learning at the edge: Real-time training on smartphone devices[J]. arXiv preprint arXiv:2105.13127, 2021.

[3] Graffieti G, Maltoni D, Pellegrini L, et al. Generative negative replay for continual learning[J]. Neural Networks, 2023, 162: 369-383.

Further, we compare neural networks, transfer learning using fine-tuning, and the proposed replay-based continuous learning to demonstrate the high-accuracy localization performance and the mitigation of catastrophic forgetting achieved by our method.

In this paper, we overcome the limitations of previous research, including the reliance on images and the focus on outdoor localization. Therefore, in this paper, we do not have reference studies for direct comparison. In future research, if similar studies apply continuous learning to improve the accuracy performance for RSS fingerprint-based indoor localization, we will conduct further analysis and comparisons.

Reviewer 2 Report

Comments and Suggestions for Authors

In the submitted manuscript, Njima and colleagues present a high-accuracy and real-time fingerprint-based continual learning localization system in a dynamic environment. In this manuscript, the authors propose a fingerprint-based Continual Learning (CL) localization system designed to retain old data while enhancing the accuracy of new data. The system works by rehearsing parameters in the lower network layers and reducing the training rate in the upper layers. Simulations conducted with fused data show that the proposed approach improves accuracy by 16 % for new data and 29 % for old data compared to TL in smaller rooms. In larger rooms, it achieves a 14 % improvement for new data and a 44 % improvement for old data over TL. This paper achieved and demonstrated a certain level of work on this point, but in my opinion, it does not have sufficient impact and novelty for publication in “Sensors”. This is largely based on the several precedents for previously reported works published in the following papers such as (i) J. Mob. Syst. Appl. Serv 2015, 1, 77. (ii) IEEE Transactions on Automation Science and Engineering 2020, 17, 1585. (iii) IEEE SENSORS JOURNAL 2022, 22, 13562. Therefore, this is a kind of extension work of these precedents applying the system to the current combination. Consequently, I conclude that this work does not meet the very high novelty and impact requirement for publication in “Sensors”. More specialized journals would be suited.

Author Response

Review 2:

In the submitted manuscript, Njima and colleagues present a high-accuracy and real-time fingerprint-based continual learning localization system in a dynamic environment. In this manuscript, the authors propose a fingerprint-based Continual Learning (CL) localization system designed to retain old data while enhancing the accuracy of new data. The system works by rehearsing parameters in the lower network layers and reducing the training rate in the upper layers. Simulations conducted with fused data show that the proposed approach improves accuracy by 16 % for new data and 29 % for old data compared to TL in smaller rooms. In larger rooms, it achieves a 14 % improvement for new data and a 44 % improvement for old data over TL. This paper achieved and demonstrated a certain level of work on this point, but in my opinion, it does not have sufficient impact and novelty for publication in “Sensors”. This is largely based on the several precedents for previously reported works published in the following papers such as (i) J. Mob. Syst. Appl. Serv 2015, 1, 77. (ii) IEEE Transactions on Automation Science and Engineering 2020, 17, 1585. (iii) IEEE SENSORS JOURNAL 2022, 22, 13562. Therefore, this is a kind of extension work of these precedents applying the system to the current combination. Consequently, I conclude that this work does not meet the very high novelty and impact requirement for publication in “Sensors”. More specialized journals would be suited.

Answer:

Thank you for the comment which is very insightful. This paper improves the localization accuracy and ameliorate problem of the catastrophic forgetting. Also we deal with the practical problem of the limitation of images because in most cases, fingerprint-based localization in reality uses RSS or CSI data rather than images.

(i) proposes an average-based algorithm using IR localization technology. It uses static anchors and mobile anchors, calculates the Euclidean Distance between the mobile node and static nodes, therefore achieves the localization by the religion prediction.

(i) is different from the one that we propose. In our paper, we: 1) utilizes machine learning and continual learning, while (i) relies on a simple average mechanism for localization by predicting the region. (i) involves less advanced artificial intelligence techniques, which may limit its applicability and potential in more complex scenarios; 2) fulfill localization in dynamic environments, while (i) focuses on the mobile anchors localization; 3) decrease the need for the number of anchors.

(ii) proposes a robust visual localization system building on top of a feature-based visual simultaneous localization and mapping algorithm and design a dynamic region detection method and use it to preprocess the input frame.

Compared to (ii), we: 1) do not use images, using RSS data in contrast which is more convenient to be observed in practical. RSS data also has less takes up less space than images; 2) predict the localization directly, while (ii) uses region detection algorithms, meaning that our proposed method is more faster and flexible without additional processing.

(iii) proposes a novel localization framework based on multiple transfer learning fusion using Generalized Policy Iteration.

Compared to (iii), our proposed method: 1) do not use images, applying RSS data literally; 2) not only improve the localization accuracy but also study the performance of catastrophic forgetting, while (iii) only focus on the localization accuracy; 3) improve the transfer learning not using transfer learning while (iii) is based on the transfer learning, which may cause catastrophic forgetting.

Based on the above, our proposed method 1) has different applied scenarios which is dynamic not static which is different with (i); 2) applies RSS data not for images data which is different with (i),(ii),(iii), which is more practical; 3) achieves not only higher localization accuracy but also avoid catastrophic forgetting which is totally different with (i),(ii),(iii). As a conclusion, our proposed method has the novelties of the practicality dealing with the limitation of images and encourages the practical fingerprint continual learning study in indoor dynamic localization field, which is the most important step in this field.

Reviewer 3 Report

Comments and Suggestions for Authors

This paper presents a new data-based CL approach for real-time indoor localization, which utilizes replay policy and reintroduces previously learned information. Accordingly, the localization accuracy is improved through leveraging features that extracted from multiple lower layers instead of relying on a single predetermined layer. The paper is well written and organized and the authors have managed to properly introduce and discuss the proposed model. Meanwhile, there are some issues that should be addressed by the authors, which are:

The authors have not provided any comparison with any related work, while in section one it has been indicated that there are many related models which utilizes CL, and especially references 34-38. The proposed model should be justified against other related approaches presented in literature to determine its significance and superiority upon other related techniques.

The authors should include a table that lists all the symbols with their definitions as it hard to comprehend some symbols used in the algorithms and especially in algorithm 3.

The CDF for localization error in Fig. 4 (a) seems to be nearly equal regarding different scenarios (error from fine tuning, error from CL with RM =0%, error from CL with RM =20%, error from CL with RM =100%). This point should be discussed by the authors and indicate the reasons behind this behavior.

Comments on the Quality of English Language

The English language seems to be fine

Author Response

Review 3:

This paper presents a new data-based CL approach for real-time indoor localization, which utilizes replay policy and reintroduces previously learned information. Accordingly, the localization accuracy is improved through leveraging features that extracted from multiple lower layers instead of relying on a single predetermined layer. The paper is well written and organized and the authors have managed to properly introduce and discuss the proposed model. Meanwhile, there are some issues that should be addressed by the authors, which are:

Comment 1: The authors have not provided any comparison with any related work, while in section one it has been indicated that there are many related models which utilizes CL, and especially references 34-38. The proposed model should be justified against other related approaches presented in literature to determine its significance and superiority upon other related techniques.

Answer:

Thank for this question which is clear. This paper improves localization accuracy and addresses the problem of catastrophic forgetting. Additionally, we tackle the practical challenge of the limitation of using images, as in most real-world scenarios, fingerprint-based localization relies on RSS or CSI data rather than images.

Compared to references [34]-[38], we: 1) apply continual learning on indoor localization not outdoor localization; 2) improve the continual learning methods; 3) make the method more practical by using RSS data rather than images data.

Firstly, we focus on the indoors environments while [34]-[38] all focus on outdoor environments which is assisted by GPS signals. Furthermore, we apply rehearsal policy over multiple layers in neural networks while [34]-[38] only replay features in one layer which cannot fully utilize the existing information, resulting in missing valuable data. Also, unlikely these references, we frozen the upper layer and train the network with full speed in rehearsal layers which not only can remember the deep information in upper layers but also update the real-time information in rehearsal layers. This reduces the training time by less time on upper layers. Finally, RSS data is more practical for existing fingerprint-based localization applications while [34]-[38] are all based on the images which results in less adaption.

Therefore, compare to other researches, this work is the first to apply continuous learning to RSS fingerprint-based indoor wireless localization. We do not have reference studies for direct comparison. Even though, we compare the performance of neural networks, transfer learning take fine-tuning as the example and the improved continual learning method. In the future, if necessary, we would make more comparisons between the proposed continual learning method and other novel research which is also based on RSS data for dynamic environments.

Comment 2: The authors should include a table that lists all the symbols with their definitions as it hard to comprehend some symbols used in the algorithms and especially in algorithm 3.

Answer:

Thank you for this comment. I agree with it therefore, I added more explanation about the applied symbols in Algorithm 3.

Modification under lines 262-273 in page 8/14: The complete pipeline of the proposed CL system is outlined as follows. In this paper, key functions are applied. Gradient descent function ∇θ L(θ) is the gradient of function L(θ) with respect to θ. θ |L are parameters chosen from L layers in one network. Also key parameters in the rehearsal process are utilized. Training data D represents the raw data RSS and according real locations for training to generate the CL model. Parameters θ include weights and bias in one networks. Batch size B is the number of receivers used per iteration. Learning rate η controls the step size taken in the direction of the gradient during each update in gradient-based optimization. Layers for fine-tune L indicates the layers that are updated during the fine-tuning process. External memory M refers to a storage mechanism used to retain previous and old RSS data. Two networks work in the whole pipeline: the old network N N and the new network N N ′. N N is trained based on the old RSS data and N N ′ is trained by the new RSS data.

Comment 3: The CDF for localization error in Fig. 4 (a) seems to be nearly equal regarding different scenarios (error from fine tuning, error from CL with RM =0%, error from CL with RM =20%, error from CL with RM =100%). This point should be discussed by the authors and indicate the reasons behind this behavior.

Answer:

Thank you for the comment which is very helpful for reader to understand the paper. Here, I made this modification to explain the figure in Chapter 4.

Modification under lines 368-372 in page 12/14: It can be noticed that the results from Fine-tuning, CL with RM = 0%, RM = 20%, RM = 50%, RM = 80% and RM = 100% are very similar, This is because, in conditions where only small changes occur in the environment—indicating that the rehearsal data is similar to the previous one, further implying that the new dataset has minimal differences from the old one—Fine-tuning and CL may achieve similar accuracy performance.

Round 2

Reviewer 2 Report

Comments and Suggestions for Authors

The Manuscript is much improved now than their previous version. It can be accepted.

Reviewer 3 Report

Comments and Suggestions for Authors

Based on all amendments that have been made to the manuscript, the authors have managed to address all the required suggestions and overcome the issues in the previous manuscript

Comments on the Quality of English Language

The English language looks to be fine.